# Characterization of PVL-Positive MRSA Isolates in Northern Bavaria, Germany over an Eight-Year Period

**DOI:** 10.3390/microorganisms11010054

**Published:** 2022-12-24

**Authors:** Tobias Szumlanski, Bernd Neumann, Ralph Bertram, Alexandra Simbeck, Renate Ziegler, Stefan Monecke, Ralf Ehricht, Wulf Schneider-Brachert, Joerg Steinmann

**Affiliations:** 1Institute for Hospital Hygiene, Medical Microbiology and Clinical Infectiology, Paracelsus Medical University, Nuremberg General Hospital, 90419 Nuremberg, Germany; 2Department of Surgery, Asklepios Hospital Barmbek, 22307 Hamburg, Germany; 3Leibniz Institute of Photonic Technology (IPHT), 07745 Jena, Germany; 4InfectoGnostics Research Campus, 07743 Jena, Germany; 5Institute for Medical Microbiology and Virology, Dresden University Hospital, 01307 Dresden, Germany; 6Institute of Physical Chemistry, Friedrich-Schiller University, 07743 Jena, Germany; 7Department of Infection Prevention and Infectious Diseases, University Hospital Regensburg, 93053 Regensburg, Germany

**Keywords:** CA-MRSA, PVL, antimicrobial resistance, travel, migration

## Abstract

Purpose: Community-acquired methicillin-resistant *Staphylococcus aureus* strains (CA-MRSA) are spread worldwide and often cause recurring and persistent infections in humans. CA-MRSA strains frequently carry Panton–Valentine leukocidin (PVL) as a distinctive virulence factor. This study investigates the molecular epidemiology, antibiotic resistance and clinical characteristics of PVL-positive MRSA strains in Northern Bavaria, Germany, isolated over an eight-year period. Methods: Strains were identified by MALDI-TOF MS and antibiotic susceptibility was tested by automated microdilution (VITEK 2) or disk diffusion. PVL-encoding genes and *mecA* were detected by PCR. MRSA clonal complexes (CC) and lineages were assigned by genotyping via DNA microarray and *spa*-typing. Results: In total, 131 PVL-positive MRSA were collected from five hospital sites between 2009 and 2016. Predominant lineages were CC8-MRSA-[IV+ACME], USA300 (27/131; 20.6%); CC30-MRSA-IV, Southwest Pacific Clone (26/131; 19.8%) and CC80-MRSA-IV (25/131; 19.1%). Other CCs were detected less frequently. Resistance against erythromycin and clindamycin was prevalent, whereas all strains were sensitive towards vancomycin and linezolid. In total, 100 cases (76.3%) were causally linked to an infection. The majority (102/131; 77.9%) of isolates were detected in skin swabs or swabs from surgical sites. Conclusions: During the sample period we found an increase in the PVL-positive MRSA lineages CC30 and CC1. Compared to less-abundant lineages CC1 or CC22, the predominant lineages CC8, CC30 and CC80 harbored a broader resistance spectrum. Furthermore, these lineages are probably associated with a travel and migration background. In the spatio-temporal setting we investigated, these were arguably drivers of diversification and change in the landscape of PVL-positive MRSA.

## 1. Introduction

*Staphylococcus aureus* is a ubiquitous pathogen causing health problems in animals and humans worldwide [1,2]. Although *S. aureus* often occurs as a commensal of the human microbiota, it also accounts for many both community-acquired and healthcare-associated infections. Antibiotic resistance is a serious issue in staphylococcal diseases. For example, in the 1960s shortly after the introduction of isoxazolyl penicillins, resistant isolates were detected, leading to the concept of methicillin-resistant *S. aureus* (MRSA) [3,4]. Based upon their epidemiology, these are further categorized as community-associated, healthcare-associated, or livestock-associated (CA-, HA-, or LA-MRSA, respectively). Due to the *mecA*, or rarely the *mecC*, gene, which encode an alternated penicillin-binding protein (PBP, also referred to as PBP2a), MRSA are resistant against methicillin and other beta-lactam antibiotics [5] that are first-line treatment for *S. aureus* infections. Thus, therapy of MRSA infections has to rely on other drugs [6,7]. However, these are potentially less effective, less well-tolerated or much more expensive, such as vancomycin [8].

*S. aureus* owes its global success to a vast array of virulence factors, including Panton–Valentine leukocidin (PVL), which is encoded by *lukF*-PV and *lukS*-PV and constitutes a pore-forming toxin that attacks leukocytes such as neutrophils and monocytes [9]. PVL has previously been associated with persistent and recurring skin and soft tissue infections (SSTI), as well as severe cases of invasive disease, such as necrotizing pneumonia [10,11]. PVL was considered to be epidemiologically linked to five predominant CA-MRSA strains [12]. According to multilocus sequence typing (MLST) based upon seven housekeeping genes [13], MRSA strains are assigned to sequence types (STs), and among STs that share alleles, at least five loci are considered to belong to the same clonal complex (CC) [14]. The epidemiology of different MRSA strains has been investigated comprehensively [15,16,17], but data regarding PVL-carrying MRSA in Germany are relatively scarce or limited to outbreak investigations [18,19,20]. We hypothesize that the distribution of MRSA strains belonging to different CCs is not static, but rather dynamic in a regional setting over several years. Hence, this retrospective study focusses on the characterization of PVL-positive MRSA isolated in Northern Bavaria, Germany, between 2009 and 2016 and sheds light on the population structure and clinical and patients’ backgrounds, including travel and migration.

## 2. Results

### 2.1. Clinical Characteristics of Collected Strains

Between 2009 and 2016, a total of 131 PVL-positive MRSA isolates were identified in five hospital sites in Northern Bavaria that participated in the study. Isolates stemmed from 79 males, 51 females and one person of undocumented sex. The individuals were less than 12 months to 95 years of age (median = 30 years). A total of 100 (76.3%) isolates were causally linked to an infection. In 25 cases (19.1%), the association with an acute infection of the isolates remained inconclusive, and for six isolates (4.6%), no information was documented. In the case of 14 isolates, patient data enabled an association with travel history (4.6%; *n* = 6) or a migration context (6.1%; *n* = 8). Most MRSA specimens were isolated from various swabs. 55.0% of isolates (*n* = 72) were obtained from skin or skin appendages, and 22.9% (*n* = 30) from surgical sites of various locations. Screening smear tests accounted for 9.2% of isolates (*n* = 12) and 5.3% (*n* = 7) stemmed from respiratory specimens. The remaining 7.6% (*n* = 10) came from other or unspecified sources. Regarding the clinical discipline, the largest proportion, with 17.6% (*n* = 23) of isolates, was sampled by the department of dermatology. Others were from paediatric surgery (9.2%; *n* = 12), hand surgery (8.4%; *n* = 11), otorhinolaryngology (7.6%; *n* = 10); plastic surgery (6.9%; *n* = 9), intensive care (6.9%; *n* = 9), unspecified surgical departments (6.1%; *n* = 8), internal medicine (6.1%; *n* = 8), trauma surgery/orthopedics, ophthalmology and general surgery (5.3%; *n* = 7 each) and gynecology and oral-maxillo-facial surgery (4.6%; *n* = 6 each). For 6.1% (*n* = 8), the origin was not documented.

### 2.2. Genotyping Results

The 131 isolates were allocated to 11 different clonal complexes (CC), with CC8 as the most common (24.4%; *n* = 32), followed by CC30 (22.9%; *n* = 30) and CC80 (19.1%; *n* = 25). Other CCs were found less frequently: CC5 (9.2%; *n* = 12), CC1 and CC772 (5.3%; *n* = 7 each), CC22 (4.6%; *n* = 6), CC59 (3.8%; *n* = 5), CC88 (2.3%; *n* = 3), as well as CC152 and CC93 (1.5%; *n* = 2 each). The distribution of CCs found in each year is depicted in Figure 1.

Whereas CC5 showed a comparatively stable proportion throughout the observation period, CC8 was detected at increasing percentage from 2009 to 2014, and then rapidly declined over the last two study years. The frequency of CC30 was low between 2009 and 2011 and increased to a stable level from 2012 on. CC80 was detected throughout the entire study period with varying numbers, except in 2013. Other CCs were identified sporadically in 2–4 study years, without apparent trends. Of all CCs of this study, only CC1 (isolated 2015 and 2016) and CC88 were distributed unevenly to a significant degree over the observation period (CC1: p = 0.003; CC88: p = 0.012).

The isolates could further be assigned to 19 distinct lineages by microarray analyses. We identified 42 different *spa*-types within the study collection. Figure 2 shows the different typing levels (CC, *spa*-type) together with information of the microarray-based differentiation on strain-level (see also Appendix A). Some of these were prominent international clonal lineages, such as USA300. Most of the USA300-like isolates belonged to *spa*-type t008 (*n* = 29) and were obtained over the entire duration of the study. Other *spa*-types within the USA300 lineage were detected at smaller numbers: t024 (*n* = 4) and singletons of t068, t1617, t2648, t304 and t4229.

Most CC30 strains belonged to t019 (*n* = 18) and were identified as closely related to CC30-MRSA-IV (PVL+) Southwest Pacific Clone. The strains of CC80 were classified as CC80-MRSA-IV (PVL+) clone and predominantly belonged to *spa*-type t044 (*n* = 20). Ten of twelve CC5-isolates were classified as CC5-MRSA-IV (PVL+), carrying SCC*mec* type IV, and were *spa*-typed as t002.

The other seven clonal complexes accounted for two to seven isolates each, a total of 32 strains comprising 17 different *spa*-types. These fell into several distinct clonal lineages: CC22-MRSA-IV, CC1-MRSA-[V/VT+fus+ccrAB1], CC59-MRSA-VT Taiwan Clone, ST772-MRSA-V Bengal Bay Clone, ST93-MRSA-IV Queensland Clone, and the CC22-MRSA-[IV+fus] Regensburg Clone (a strain previously identified in a large outbreak in Bavaria).

### 2.3. Isolates Associated with Migration and Travel

Eight isolates were obtained from patients with a migration background. Three of the CC1-MRSA-[V/VT+fus+ccrAB1] isolates were demonstrably obtained from patients being asylum seekers in Germany, with one of them coming from Armenia, another one from Ethiopia and yet another one from an unclear origin. Four isolates belonging to CC80-MRSA-IV were obtained from patients from Somalia (*n* = 1), Syria (*n* = 2), and one of unknown origin, respectively. One CC88-MRSA-IV was isolated from a patient originating from Afghanistan.

Another six isolates were obtained from patients with travel history. These isolates belonged to different clonal complexes (CC5, CC8, CC30 and CC772), and different lineages: CC8-MRSA-[IV+ACME] USA300 (*n* = 3); CC5-MRSA-IV (*n* = 1), CC30-MRSA-IV Southwest Pacific Clone (*n* = 1) and ST772-MRSA-V Bengal Bay Clone (*n* = 1). Three cases of CC8-MRSA-[IV+ACME] USA300 infection were linked to U.S. military forces stationed in Germany. The CC30 isolate was obtained from a patient who had been bitten by a dog while travelling in the Philippines. Two cases of infection with the CC5-MRSA-IV and ST772-MRSA-V Bengal Bay Clone, respectively, could be traced back to insect bites in Sri Lanka, culminating in cellulitis and abscess formation. Figure 3 shows a geographical map illustrating the putative origins of isolates associated with migration and travel in this study.

### 2.4. Antibiotic Susceptibility

Susceptibility to antibiotics was assessed for 125 of 131 MRSA isolates. In six isolates, resistance could not be determined due to technical complications. As expected for MRSA, all isolates were resistant to penicillin and oxacillin/flucloxacillin, and this was also confirmed by detection of the *mecA* gene. None of the isolates showed resistance to last-line antibiotics vancomycin and linezolid.

Data shown in Table 1 underline that the distinct genotypes are associated with different resistance patterns. In total, 38.9% of the isolates were sensitive towards non-beta-lactam antibiotics. When tested against up to four additional antibiotics, 61.1% were resistant to at least one, 34.4% to more than one, 14.5% to more than two, and 2.3% were resistant to all tested antibiotics. The lineage CC772 showed most resistance, with all but one isolates resistant to at least two non-beta-lactam antibiotics. Isolates of CC93 were susceptible to all tested antibiotics (except penicillin and oxacillin/flucloxacillin). The predominant lineages in the study collection, CC8, CC30, and CC80, were harboring more resistance than the low-abundant CC1 or CC22. In addition, most isolates that were obtained from patients with travel and migration background belonged to these resistant lineages, as well as to CC772.

## 3. Discussion

The present study characterizes 131 isolates PVL-positive MRSA collected in five hospitals in Northern Bavaria, Germany, between 2009 and 2016. All of these isolates carried either SCC*mec* type IV or V/V_T_, which is consistent with the association of PVL and CA-MRSA observed in other studies [21].

The three most prevalent clonal complexes in our study were CC8, CC30, and CC80. This is in accordance to prevalence in Germany in 2015 and 2016 with CC8 ranked first, CC30 ranked second, and CC80 ranked fourth [22]. In Germany, CC80 showed a very high rate of isolates positive for PVL [23]. The high number of isolates belonging to CC8, USA300 we found is also consistent with an international study from 2013, in which CC8 was one of the two most common types of PVL-positive MRSA [24]. Among the major three CCs in our study, no preference for a specific anatomic site of isolation could be identified. CC30, the abundance of which increased in the latest two years of our study, is associated with patients with travel history or a migration background from/to Thailand, Mexico, Dominican Republic, Serbia, Afghanistan, Syria and Somalia [22]. In one carrier of CC30, infection could be traced back to a dog bite at the Philippines, which is also in agreement to a spread of the strain in the Asia-Pacific regions [17]. Patients from Somalia, Syria, and one of unknown origin also carried CC80 strains in our study, which reflects the distribution of this CC in the Middle East and parts of Africa [16]. CC5, found 12-times in our study, ranged among the predominant MRSA lineages in Germany between 2000 and 2010 [25] and also in 2015 and 2016 [22]. One CC5 isolate in our study was assigned to Sri Lanka, in which CC5 is prevalent [26]. CC772 is also known as a Bengal clone, first reported from India and later identified in the Bengal Bay area [27]. One CC772 isolate from our study also originated from Sri Lanka. The Bengal Bay Clone variant CC772-MRSA-[pseudoSCCmec class C] (PVL+) that we detected represents a novel lineage not previously described in Germany.

Three isolates of CC88 were identified in our study. In 2015 and 2016, 6.3% of CA-MRSA sent to the German National Reference Center for Staphylococci belonged to CC88 and were associated with patients from Iraq, Iran and Afghanistan [22]. Accordingly, one of our CC88 isolates was obtained from an Afghan person.

CA-MRSA isolates of CC1 were reported in association with patients with travels to or from Syria, Iraq, and Eritrea, with nearly half of the isolates being positive for PVL. Furthermore, from 2013 onwards, clusters with PVL-positive CC1-t127, one of the two *spa* types present in CC1 isolates in our study, have regularly been isolated in German hospitals [22]. It has been reported that a multidrug-resistant CC1 clone emerging in Germany likely originated in South-Eastern Europe [28]. In our study, CC1 isolates were detected in 2015 (*n* = 3) and in 2016 (*n* = 4). Three of the isolates could be traced back to asylum seekers in association with the refugee surge in 2015, when more than one million people sought asylum in Germany [29,30]. A high percentage of refugees originated from Syria, Iraq, Afghanistan, and the African continent. A Dutch study reported that up to 70% of PVL-positive MRSA examined had been retrieved from refugees from Eritrea and Syria, suggesting a high prevalence among these populations [31]. Furthermore, most of these isolates from the Eritrean refugees belonged to CC1, most of which carried *spa*-type 5100, one of the two *spa* types found in CC1 in our study. Similarly, the 2015 report of the Statens Serum Institute in Denmark recorded an outbreak of t5100-MRSA in an asylum centre [32]. In a recent study which analyzed MRSA obtained from refugees in Germany, the authors found predominantly *spa*-types t021, t084, t304, t991 and t4983, and identified t304 and t991 as local types for the Middle East [33]. In contrast, none of these genotypes was detected in our study associated with travel or migration of the patients. This is because they are associated with CC6 and CC913 that are usually not linked to the carriage of PVL genes. It was shown that the global CA-MRSA landscape is dynamic and strongly influenced by travel and migration [15].

Besides their innate resistance to beta-lactam antibiotics, PVL-positive MRSA frequently avail of further resistance factors [24]. Almost 25% of isolates in our study were resistant to clindamycin. Lower resistance rates were reported from another region in Southwestern Germany, with a portion of 15.2% of resistant isolates [23]. By contrast, resistance against linezolid was not detected in any of the strains in our study, suggesting that this drug remains an option for treating PVL-positive MRSA infections. The highest observed resistance rate for non-beta-lactam antibiotics was against erythromycin, with more than 38% of non-susceptible isolates, followed by the fluoroquinolone moxifloxacin with a resistance rate of nearly 30%. These results are in accordance with other findings from Europe and Southern Germany that reported resistance rates of about 39–40% for erythromycin and of about 24–25% for ciprofloxacin, another fluoroquinolone, [23,24].

Resistance rates for gentamicin (15.2%) and trimethoprim-sulfamethoxazol (8.8%) observed in our study were comparable to those of two studies in Heidelberg, Southwestern Germany [23,34], whereas 17.6% of our isolates displayed phenotypic resistance against tetracycline, compared to 39.1% reported there.

The prevalence of PVL in MRSA in Germany was reported as 1% to 2.7% between 2004 and 2011 [35], as well as 1.8% in 2005 and 3.1% in 2006 [36]. Between 2015 and 2018 a total of 6.2% PVL-positive MRSA strains were isolated from hospitalized patients [23]. Systematic screening and reporting of PVL in *S. aureus* isolates are neither mandatory nor well established in Germany. According to federal law, reporting is obligatory only for MRSA from blood or cerebrospinal fluid, regardless of the presence or absence of PVL. Therefore, the decision to test for PVL in a clinical MRSA isolate is mostly in the hands of the physician and is taken based on criteria which may vary between hospitals and laboratories. Hence, it can be suspected that several PVL-positive MRSA may have been missed in this study. Furthermore, since clinical data were not fully available for all patients of this study, thus possible clusters or outbreaks may go unnoticed and the true prevalence of certain lineages in the overall population be over- or underrepresented. Only in two cases, transmission between family members could be established, but no indication of larger outbreaks within the community or the hospital could be found.

## 4. Conclusions

The present study provides insights into the epidemiology and antibiotic resistance of PVL-positive CA-MRSA isolates between 2009 and 2016 in Northern Bavaria. According to our data, migration and international travel contributed to diversification and change in the PVL-positive MRSA landscape of in this region. Monitoring the landscape of different MRSA lineages by means of molecular strain-typing [37] can aid in both the prevention of transmission and in therapy. Our results suggest that implementing surveillance strategies for PVL-MRSA would contribute to extending our currently fragmentary knowledge of this important opportunistic pathogen’s distribution.

## 5. Materials and Methods

### 5.1. Ethical Review and Approval

This study was registered and authorized at the Study Center of Nuremberg General Hospital (SZ_W_090.18-H). An ethics statement was obtained from the Institutional Review Board of the Paracelsus Medical University in Nuremberg (IRB-2022-006).

### 5.2. Sampling of Isolates

Between 2009 and 2016, MRSA isolates were collected from routine clinical samples in five regional hospitals in Northern Bavaria, Germany. Isolates were identified as *S. aureus* by MALDI-TOF MS (Bruker, Bremen, Germany) and investigated for resistance and virulence characteristics. Anonymized clinical and patient metadata, such as anatomic isolation sites, indication of possible infection linked to the respective isolate as well as possible travel and migration history, were analyzed retrospectively.

### 5.3. PCR Screening for Resistance Genes and PVL-Genes

A total of 166 isolates of *S. aureus* isolates were analyzed for the presence of *mecA* and PVL-genes by the “GenoType MRSA” PCR assay (Hain Lifescience GmbH, Nehren, Germany), according to the manufacturer’s instructions. Of these, 131 carried both *mecA* and PVL. The remaining 35 strains were regarded as PVL-positive MSSA strains.

### 5.4. Antibiotic Susceptibility Testing

Susceptibility to linezolid, vancomycin, moxifloxacin, clindamycin, erythromycin, trimethoprim-sulfamethoxazol, tetra-/doxycyclin, gentamicin, and (flucl-)oxacillin was tested by either the agar diffusion method or using the Vitek 2 system (bioMérieux, Marcy-l’Etoile, France) in 125 of 131 isolates. Susceptibilities were interpreted according to the version 12.0 of released by the European Committee on Antibiotic Susceptibility Testing [38].

### 5.5. Microarray Analyses and Spa-Typing

Isolates positive for *mecA* and PVL were subjected to genotyping by DNA microarray and *spa*-typing. DNA was extracted from freshly streaked colonies using the “DNeasy Blood & Tissue Kit” (Qiagen GmbH, Hilden, Germany). For *spa*-typing, a PCR amplification and Sanger-sequencing was conducted as described [39]. Briefly, the “*S. aureus* Genotyping Kit 2.0” (Alere GmbH, Jena, Germany) was used to conduct DNA microarray analysis. It contains markers for a large quantity of virulence determinants, including genes for SCCmec typing (see Appendix A). Amplified isolates’ DNA was biotin-labeled and hybridized on the oligonucleotide DNA array as described previously [40]. Data were analyzed using the ArrayMate Reader (Alere GmbH, Jena, Germany) and the included software, and visualized by the RAWGraphs 2.0 web-tool [41].

### 5.6. Statistics

To statistically assess the occurrence of clonal CCs per year, discrete uniform distribution-tests, Chi-squared tests and Monte Carlo simulations were applied. The null hypothesis was that each CC of our sample was evenly distributed during the observation period. P-values below 0.05 indicated significant non-even distribution and rejection of the null hypothesis. Statistical analyses were performed using Microsoft Excel 365 (version 2112) and Mathematica (version 13), Wolfram Research Inc., Champaign, IL, USA.

## Figures and Tables

**Figure 1 microorganisms-11-00054-f001:**
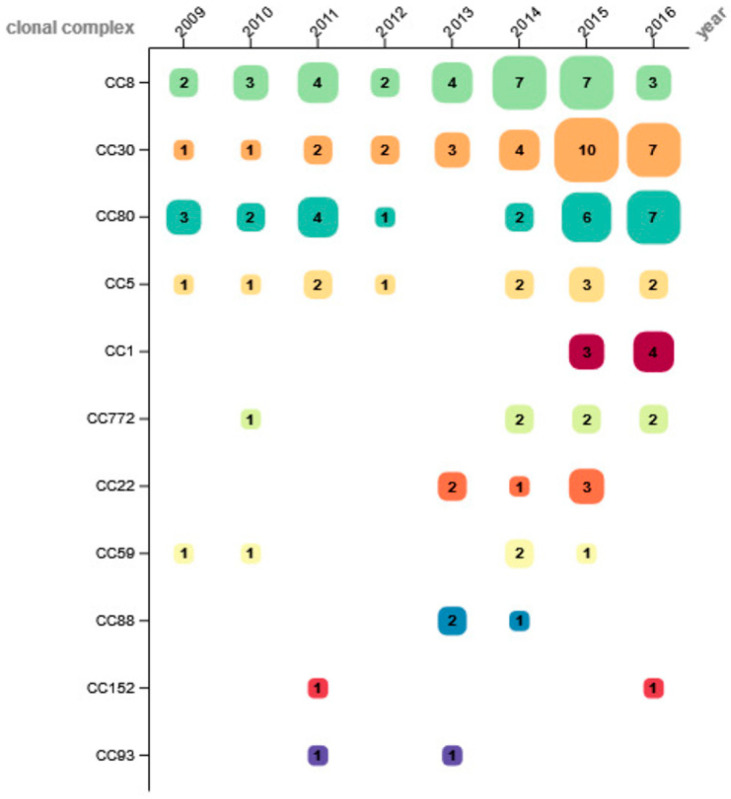
Number of determined clonal complexes per year.

**Figure 2 microorganisms-11-00054-f002:**
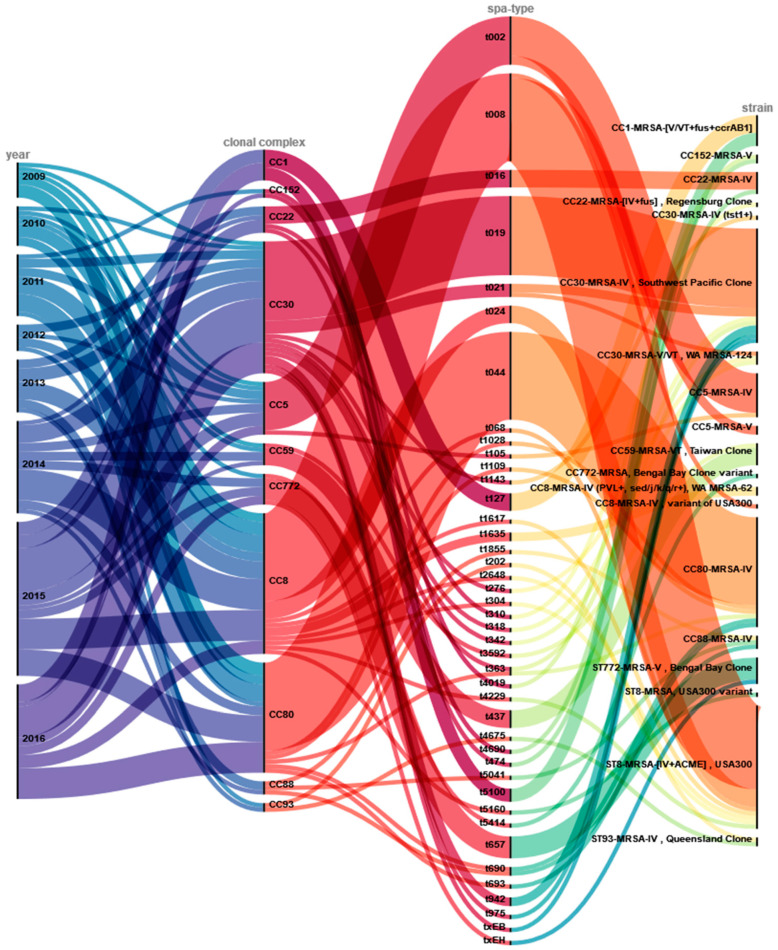
Overview of genotyping results and assignment to clonal lineages. This alluvial diagram depicts the year of isolation (year) and the distributions of genotypes (clonal complex; *spa*-type), as well as microarray-based strain assignments (strain). Colored lines visualize and link the allocation of these results.

**Figure 3 microorganisms-11-00054-f003:**
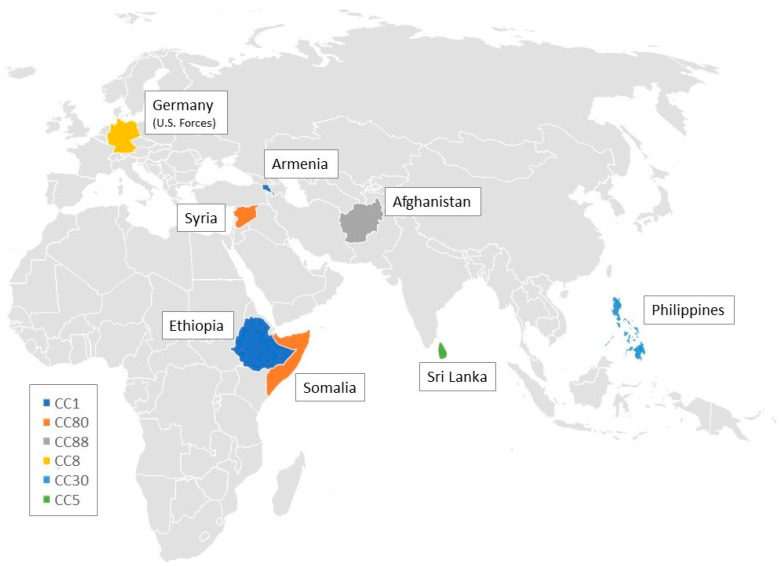
Geographical map illustrating the origins of clonal complexes associated with travel and migration in this study.

**Table 1 microorganisms-11-00054-t001:** Overview of resistances (in %) to antibiotics for each clonal complex and overall resistance rates. The upper line denotes the number of the clonal complex (CC).

	1	152	22	30	5	59	772	8	80	88	93	Total
Linezolid	0	0	0	0	0	0	0	0	0	0	0	0
Vancomycin	0	0	0	0	0	0	0	0	0	0	0	0
Moxifloxacin	0	0	0	11.1	0	20	100	37.5	4	0	0	15.5
Clindamycin	0	0	0	7.1	25	60	28.6	31	36	33.3	0	23.3
Erythromycin	0	0	0	14	33.3	60	57.1	75.9	40	33.3	0	38.4
Trimethoprim-Sulfamethoxazol	0	50	0	21.4	0	0	57.1	0	0	0	0	8.8
Tetra-/Doxycyclin	50	50	0	10.7	0	20	0	0	52	33.3	0	17.6
Gentamicin	100	50	0	0	33.3	0	85.7	6.9	0	0	0	15.2
(Flucl-) Oxacillin	100	100	100	100	100	100	100	100	100	100	100	100

## Data Availability

The datasets generated for this study are available on request to the corresponding author.

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
