# Peer review of "Characterization of PVL-Positive MRSA Isolates in Northern Bavaria, Germany over an Eight-Year Period"

_microorganisms, 2022, doi:10.3390/microorganisms11010054_

Round 1
Reviewer 1 Report
The current review is of strong interest in that it is a historical overview. Such work is relatively rare and merits encouragement.
All is well until one hits the figures and then woe is me. None of the figures is usable, not due to scientific use which is fine but due to them being illisible. To reconsider the authors must make the effort to ensure the reader can make sense of the data in the figures without spending a large amount of effort re-scaling etc the figures.
I like the work but figures are a key and must be usable.
Revise as requested
Author Response
The current review is of strong interest in that it is a historical overview. Such work is relatively rare and merits encouragement.
All is well until one hits the figures and then woe is me. None of the figures is usable, not due to scientific use which is fine but due to them being illisible. To reconsider the authors must make the effort to ensure the reader can make sense of the data in the figures without spending a large amount of effort re-scaling etc the figures.
I like the work but figures are a key and must be usable.
Revise as requested
--> We thank the reviewer for generally responding favourably to our manuscript. We have deleted Fig. 1 and have described its data in the text. While we unfortunately cannot increase font sizes in Fig. 3 (now Fig. 2) due to spatial constraint, we have done so in Fig. 2 (now Fig. 1) to improve readability.
Reviewer 2 Report
The manuscript titled “Characterization of PVL-positive MRSA isolates in Northern Bavaria, Germany over an eight-year period” written by Szumlanski et al. focuses on the molecular epidemiology, antibiotic resistance as well as clinical characteristics of PVL-positive MRSA strains in Northern Bavaria. The study concerns to 131 PVL-positive MRSA collected from five hospital sites between 2009 and 2016. Since, MRSA is currently considered among the most relavant pathogen responsible of serious chronic infections very difficul to treat with conventional antibiotics, studies on its antibiotic resistance mechanisms are strongly required.
The manuscript can be accepted for the publication after minor revisions:
- Introduction: relevant recent references on the antibiotic resistance of MRSA should be added, e.g. ChemMedChem, 2021, 16(1), pp. 65–80 and Future Medicinal Chemistry, 2021, 13(6), 529–531
- Figure 1: is not readable, increase the font of the charachter
- Table 1: adapt column size to the writing
Author Response
...
The manuscript can be accepted for the publication after minor revisions:
- Introduction: relevant recent references on the antibiotic resistance of MRSA should be added, e.g. ChemMedChem, 2021, 16(1), pp. 65–80 and Future Medicinal Chemistry, 2021, 13(6), 529–531
-->These references were added to the Introduction section.
- Figure 1: is not readable, increase the font of the character
--> Figure 1 has been deleted; the relevant information was added to the main text to a table.
- Table 1: adapt column size to the writing
--> This will be accomplished during the final layout by the publisher.
Reviewer 3 Report
These are my main comments on the manuscript (microorganisms-2074900) entitled “Characterization of PVL-positive MRSA isolates in Northern Bavaria, Germany over an eight-year period”. The study investigates the molecular epidemiology, antibiotic resistance as well as clinical characteristics of PVL-positive MRSA. Following substantial revisions should be incorporated in the manuscript prior to acceptance.
1. I have concerns about the manuscript sections that I believe need to be addressed in order to improve its clarity.
2. A hypothesis for this work is needed.
3. In methods, authors should explain the statistical methods used in each experiment.
4. In results, statistical methods are missing. The authors must present the values (F-values, degree freedom, p-value, etc.) obtained in each statistical analysis.
5. Other revisions could be checked in PDF attached.

Author Response
...
Following substantial revisions should be incorporated in the manuscript prior to acceptance.
- I have concerns about the manuscript sections that I believe need to be addressed in order to improve its clarity.
--> We have followed the order of sections as proposed of “Microorganisms”. As for the subheadings in the results section, we have rephrased the first one to improve clarity.
- A hypothesis for this work is needed.
--> We have added a hypothesis at the end of the Introduction section.
- In methods, authors should explain the statistical methods used in each experiment.
--> We have added information about statistic analyses in the Materials and Methods section.
- In results, statistical methods are missing. The authors must present the values (F-values, degree freedom, p-value, etc.) obtained in each statistical analysis.
--> We now also describe our statistical findings in the results section.
- Other revisions could be checked in PDF attached.
--> We thank the reviewer for having gone detailed through the manuscript regarding orthography and hinting at flaws and mistakes. We have made changes as suggested. As for mecA detection by PCR (see Abstract) we feel that it does not require explanation.
Reviewer 4 Report
The study is devoted to the important problem of the spread of MRSA strains carrying the Panton-Valentine leucocidin genes among patients in hospitals in Bavaria.
The study is presented in the traditional manner for this type of works: the strains were obtained from different patients, the presence of mec-genes was established, DNA microarray and spa-typing of MRSA, as well as analysis of their antibiotics sensitivity were made.
Thus, the work contains reliable data that can eventually be used for extensive analysis. However, so far it is only an element of a common puzzle, that has not yet been assembled.
The main problem of this work is limited sampling.
Line 244: “A total of 166 strains were analyzed for the presence of mecA and PVL-genes…”
Were only 166 patients with problems caused by Staphylococcus aureus in five hospitals in 8-year period?
Does it mean that 35 strains that did not carry pvl-genes were MRSA? And what is known about MSSA from patients in these hospitals at that time?
If it is possible to supply data on the distribution of pvl-positive HA-MRSA strains in the same hospitals for the same period of time?
The second vulnerability is the lack of information for the epidemiological analysis.
The authors emphasize the importance of the migration and the travel factor for the spread of various clonal lines, however, the number of patients with a migration past and travel experience is too small, their geographical locations are too disjointed and we have little factual information about how long they arrived from countries before taking swabs to these the data could be regarded as convincing.
The authors themselves point to possibly underestimated results - due to the non-systematic detection of PVL-strains, the lack of complete information about patients, etc. In this case, this should be compensated by a broader analysis of strains (including MSSA or NA-MRSA, or PVL-negative in the same places at the same time) or by using data from other territories.
And finally, I see a lot of experimental data (Table in the supporting material), but there are not even attempts to statistically analyze them.
Without this, the conclusions do not look completely convincing, although the results are consistent with the general trends presented in a huge number of similar works.
For example, in this work (doi: 10.1038 / s41598-021-95115-2), 1761 strains were studied for 11 years in one medical center, patients were clustered and the data obtained were statistically processed.
Line 285 REFERENCES
It seems to me that the authors should use the results of these works in the discussion
https://doi.org/10.1016 / j.meegid.2019.01.021
https://doi.org/10.1093/jac/dkab341
https://doi.org/10.3389/fmicb.2018.01485
https://doi.org/10.1128/mSphere.00226-20
I would like to note Figure 2 as a very successful way of graphically displaying data, where we can see – characteristic clonal lines, in relation to which others can be considered as rare or random. However, more data is needed for such conclusions.
But Figure 3, although bright and unusual, is too difficult to perceive.
It might be advisable to use a geographical map to present the results – on which the distribution areas of certain clonal lines would be indicated in color, patients (migrants and travelers) can also be indicated on such a map, so that in this work they are calculated in units.
Line 46
It would be better “in the 1960s” instead of “…1960ies”
Line 193 ca-MRSA
“community-associated” should be written in capital letters
Line 206
unnecessary comma at the end of the line
There have been great many works of this design in the literature over the past decade, and they are undoubtedly needed, given the importance of the problem of the spread of resistant forms, however, I would like to see not just a presentation of data, but a really reliable analysis that could point to the causes and dynamics of MRSA.
Author Response
...
The main problem of this work is limited sampling.
Line 244: “A total of 166 strains were analyzed for the presence of mecA and PVL-genes…”
Were only 166 patients with problems caused by Staphylococcus aureus in five hospitals in 8-year period?
--> Screening for MRSA or PVL-positive S. aureus strains is not mandatory in Germany. The amount of 166 isolates represents a random sample.
Does it mean that 35 strains that did not carry pvl-genes were MRSA? And what is known about MSSA from patients in these hospitals at that time?
--> The remaining 35 strains can be regarded as PVL-positive MSSA strains. We have added this sentence to the Materials and Methods section. Since we focussed on MRSA strains only, we did not follow up on these isolates.
If it is possible to supply data on the distribution of pvl-positive HA-MRSA strains in the same hospitals for the same period of time?
--> Unfortunately, these data are not available, due to the facultative nature of MRSA screening in German hospitals (see above).
The second vulnerability is the lack of information for the epidemiological analysis.
The authors emphasize the importance of the migration and the travel factor for the spread of various clonal lines, however, the number of patients with a migration past and travel experience is too small, their geographical locations are too disjointed and we have little factual information about how long they arrived from countries before taking swabs to these the data could be regarded as convincing.
--> We agree that we may have overstated these aspects based upon a rather limited amount of data in this regard. Therefore, key sentences and paragraphs were rephrased.
The authors themselves point to possibly underestimated results - due to the non-systematic detection of PVL-strains, the lack of complete information about patients, etc. In this case, this should be compensated by a broader analysis of strains (including MSSA or NA-MRSA, or PVL-negative in the same places at the same time) or by using data from other territories.
--> Since this manuscript focuses on the Northern Bavarian region, comparisons with other regions must be carefully interpreted. However, we refer to a number of other studies about MRSA epidemiology in Germany.
And finally, I see a lot of experimental data (Table in the supporting material), but there are not even attempts to statistically analyze them.
Without this, the conclusions do not look completely convincing, although the results are consistent with the general trends presented in a huge number of similar works.
--> We thank the reviewer for the suggestion to take a deeper look into statistics. In this revised version we present p-values calculated by Chi-squared tests and Monte Carlo simulation in order to assess whether CC distribution over time is random or not. We find two CCs with significantly uneven distribution in the observation period and highlight these in the Results and Discussions sections.
For example, in this work (doi: 10.1038 / s41598-021-95115-2), 1761 strains were studied for 11 years in one medical center, patients were clustered and the data obtained were statistically processed.
--> We thank the reviewer for this valuable resource and cite this in the Conclusions section.
Line 285 REFERENCES
It seems to me that the authors should use the results of these works in the discussion
https://doi.org/10.1016 / j.meegid.2019.01.021
https://doi.org/10.1093/jac/dkab341
https://doi.org/10.3389/fmicb.2018.01485
https://doi.org/10.1128/mSphere.00226-20
-->Two of these papers are now cited at appropriate places in the Discussion section. We reckon that Earls et al. Front. Microbiol. 2021 is not too closely related to our study and that Baig et al. mSphere 2020 may also be omitted, since CC152 did not play a major role in our study.
I would like to note Figure 2 as a very successful way of graphically displaying data, where we can see – characteristic clonal lines, in relation to which others can be considered as rare or random. However, more data is needed for such conclusions.
But Figure 3, although bright and unusual, is too difficult to perceive.
--> This Figure is certainly complex, but it also contains a lot of information about the different characteristics of the investigated isolates. We would prefer to leave it in the manuscript and hope the reviewer will agree.
It might be advisable to use a geographical map to present the results – on which the distribution areas of certain clonal lines would be indicated in color, patients (migrants and travelers) can also be indicated on such a map, so that in this work they are calculated in units.
--> We thank the reviewer for this suggestion and have now added a map.
Line 46
It would be better “in the 1960s” instead of “…1960ies”
--> corrected.
Line 193 ca-MRSA
“community-associated” should be written in capital letters
--> corrected.
Line 206
unnecessary comma at the end of the line
--> corrected.
Round 2
Reviewer 1 Report
Much improved - the figures now aid the reader
Accept as is
Reviewer 4 Report
I am satisfied with the answers of the authors and the corrections made.